# Evaluation of the Ginsburg Scheme: Where Is Significant Prostate Cancer Missed?

**DOI:** 10.3390/cancers13102502

**Published:** 2021-05-20

**Authors:** August Sigle, Cordula A. Jilg, Timur H. Kuru, Nadine Binder, Jakob Michaelis, Markus Grabbert, Wolfgang Schultze-Seemann, Arkadiusz Miernik, Christian Gratzke, Matthias Benndorf, Rodrigo Suarez-Ibarrola

**Affiliations:** 1Department of Urology, Faculty of Medicine, University of Freiburg—Medical Centre, 79110 Freiburg, Germany; august.sigle@uniklinik-freiburg.de (A.S.); cordula.jilg@uniklinik-freiburg.de (C.A.J.); jakob.michaelis@uniklinik-freiburg.de (J.M.); markus.grabbert@uniklinik-freiburg.de (M.G.); wolfgang.schultze-seemann@uniklinik-freiburg.de (W.S.-S.); arkadiusz.miernik@uniklinik-freiburg.de (A.M.); christian.gratzke@uniklinik-freiburg.de (C.G.); 2Urologie am Ebertplatz, 50668 Cologne, Germany; thkuru@gmail.com; 3Institute of Digitalization in Medicine, Faculty of Medicine and Medical Center, University of Freiburg, 79110 Freiburg, Germany; nadine.binder@uniklinik-freiburg.de; 4Department of Radiology, Faculty of Medicine, University of Freiburg—Medical Centre, 79110 Freiburg, Germany; matthias.benndorf@uniklinik-freiburg.de

**Keywords:** prostatic neoplasms [MeSH], image-guided biopsy [MeSH], fusion biopsy, Ginsburg scheme, biopsy strategy

## Abstract

**Simple Summary:**

Systematic biopsy according to the Ginsburg scheme is widely used to complement MRI-targeted biopsy for the diagnosis of prostate cancer. This is the first study to evaluate the distribution of cancerous lesions that were missed by the Ginsburg scheme. We found that significant prostate cancer lesions were missed in 3.6% of patients. The majority of the missed lesions (17/39, 43.6%) were localized within the anterior region of the prostate. Complementing the Ginsburg scheme by adding biopsy cores to this region may be considered in certain patients who were extensively pre-biopsied without a conclusive diagnosis or when targeted biopsy was not possible. Moreover, based on Ginsburg scheme sectors and newly defined blind sectors, we developed a new sector map of the prostate that can be applied to focal therapy planning and for the follow-up management of patients under active surveillance.

**Abstract:**

Background: Systematic biopsy (SB) according to the Ginsburg scheme (GBS) is widely used to complement MRI-targeted biopsy (MR-TB) for optimizing the diagnosis of clinically significant prostate cancer (sPCa). Knowledge of the GBS’s blind sectors where sPCa is missed is crucial to improve biopsy strategies. Methods: We analyzed cancer detection rates in 1084 patients that underwent MR-TB and SB. Cancerous lesions that were missed or underestimated by GBS were re-localized onto a prostate map encompassing Ginsburg sectors and blind-sectors (anterior, central, basodorsal and basoventral). Logistic regression analysis (LRA) and prostatic configuration analysis were applied to identify predictors for missing sPCa with the GBS. Results: GBS missed sPCa in 39 patients (39/1084, 3.6%). In 27 cases (27/39, 69.2%), sPCa was missed within a blind sector, with 17/39 lesions localized in the anterior region (43.6%). Neither LRA nor prostatic configuration analysis identified predictors for missing sPCa with the GBS. Conclusions: This is the first study to analyze the distribution of sPCa missed by the GBS. GBS misses sPCa in few men only, with the majority localized in the anterior region. Adding blind sectors to GBS defined a new sector map of the prostate suited for reporting histopathological biopsy results.

## 1. Introduction

In prostate cancer (PCa) diagnosis, adding a synchronous systematic biopsy (SB) to a multiparametric magnetic resonance imaging-targeted biopsy (MR-TB) is the current standard of care [1,2,3]. Transperineal biopsy (TP) offers the advantage of avoiding the rectal flora being inoculated into the bloodstream and has, therefore, gained increasing popularity [4]. TP systematic biopsies have typically been performed according to the 20-zone Barzell-Melamed template [5,6,7]. However, following a 2013 consensus, the Ginsburg scheme (GBS) was proposed to standardize TP systematic biopsies and to encourage prospective studies and multi-center collaborative data analysis [8]. However, the GBS mainly targets the peripheral zone and systematically omits particular sectors of the prostate. There are scarce studies reporting cancer detection rates by MR-TB and SB according to GBS that allow for an isolated analysis of each procedure [9,10]. To the best of our knowledge, this is the first study to analyze the distribution of PCa that was missed by the GBS and diagnosed by MR-TB only. This is relevant for patients, for example, in whom PCa is suspected but cannot undergo MRI or when MRI does not reveal any suspicious lesions.

The study’s primary aim was to evaluate where significant PCa (sPCa) is missed by the GBS based on the re-localization of cancerous lesions that were detected by MR-TB only. The secondary aim was to identify factors associated with sPCa being missed or underestimated by the GBS.

## 2. Materials and Methods 

### 2.1. Patient Cohort

We analyzed a retrospective single-center cohort of patients that underwent robot-assisted TP of the prostate with both MR-TB plus synchronous SB according to the GBS between October 2015 and December 2019. The indication for prostate biopsy was based on elevated prostate specific antigen (PSA), abnormal digital rectal examination or suspicious findings in multiparametric MRI (mpMRI) or as part of the re-biopsy routine in men under active surveillance. Exclusion criteria were incomplete clinical data and use of a systematic biopsy scheme deviating from the GBS.

### 2.2. Biopsy Procedure

Robot-assisted mpMRI/TRUS fusion biopsy of the prostate (iSRobot Mona LisaTM^®^, Biobot Surgical, Singapore, Singapore) was performed as a combined strategy by targeting suspicious lesions followed by SB in the same session. Procedural details were described in previous investigations [11]. Eight different surgeons performed the biopsy procedures that were reviewed for this study. In the majority of cases the same surgeon performed the planning and execution of both MR-TB and SB. For systematic biopsy planning, the GBS was applied [8]. The number of cores taken per Ginsburg sector varied between patients and depended on prostate volume. The procedure was performed in lithotomy position under general anesthesia. Antibiotic prophylaxis or local anesthesia were not administered.

### 2.3. Data Collection and Statistical Analysis

Clinical data were extracted by reviewing patients’ electronic medical records. Baseline characteristics included age, history of previous biopsy, active surveillance status, PSA, prostate volume by MRI, number of total biopsy cores, random biopsy cores and target biopsy cores, mpMRI findings according to the Prostate Imaging: Reporting and Data System (PI-RADS), number of suspicious lesions and histopathological prostate biopsy findings according to the classification of the International Society of Urological Pathology (ISUP). sPCa was defined in accordance with the definition applied by the PI-RADS lexicon as the presence of any cancer graded as ISUP 2 or higher and/or dependent on cancer core length considering ≥5 mm as clinically significant [12].

Continuous variables are described as median with interquartile range (IQR) acknowledging that the data is non-normally distributed. Categorical variables are described with integers and percentages. For evaluating any difference in the distribution of continuous variables between comparison groups, Mann-Whitney U test was performed. Distributions of categorical variables between comparison groups and the distribution of cancerous lesions across different sectors were investigated using Pearson’s chi-squared test. When observed frequencies per cell were less than five, we used Fisher’s Exact test. Univariate and multivariate regression analysis were performed for the identification of predictors for sPCa solely localized within a blind sector and sPCa localized specifically within the AR. Stepwise backward elimination with Akaike information criterion (AIC) stopping criterion was used to select an important subset of covariates. *p*-value < 0.05 was considered statistically significant. SPSS^©^ statistics 27 (IBM, Armonk, New York, NY, USA) and R version 4.0.3 [13] were used for statistical analysis. Slicer version 4.10.2 was used for the visualization of prostate maps [14].

### 2.4. Evaluation of the Ginsburg Scheme

The performance of the GBS was evaluated with the histopathological reference standard of a combined biopsy. We defined four inherent blind sectors of the GBS: The anterior region (AR), the central transition zone (CTZ), and two basal sectors that were localized basal to the anterior respectively posterior core group of the GBS—basoventral (BV) and basodorsal (BD). Figure 1 shows the Ginsburg sectors (also referred to as non-blind sectors) and the newly defined blind sectors. For the subset of patients in whom sPCa was solely diagnosed by MR-TB or upgraded from non-significant PCa (nsPCa) to sPCa by MR-TB, cancerous target lesions were re-localized onto a newly defined sector map. For this purpose, stored biopsy plans were reviewed in the biopsy software (Uro-Biopsy version 4.2.0, Biobot Surgical, Singapore) and cancerous target cores were identified with respect to the histopathological report. Since the reports did not contain any directional information, we referred to the center of the biopsy core for the re-localization in craniocaudal direction. In the case of more than one cancerous target lesion per patient, the region with first higher ISUP grade and second higher cancer core length was considered as index lesion for patient-level analysis.

### 2.5. Analysis of Prostatic Configuration

For quantitative analysis of the glands’ configuration, we applied the previously described presumed circle area ratio (PCAR) index. The PCAR was originally developed to describe the transformation of hyperplastic prostates from an initial ellipsoid shape to a less oval and more rounded contour [15]. The PCAR index of the cohort where sPCa was missed within the AR only was compared to a sub-cohort constructed by means of a 2:1 nearest neighbor matching on the propensity score accounting for age and PSA value. For statistical analysis, Welch’s two sample test was applied. The evaluation was performed blinded to the biopsy results.

## 3. Results 

### 3.1. Study Cohort

Overall, 1087 consecutive patients underwent robot-assisted TP with both MR-TB plus synchronous SB according to GBS at our center. Three men were excluded due to variations in the GBS, resulting in a total of 1084 analyzed patients.

### 3.2. Baseline Characteristics

Table 1 presents the baseline characteristics of the total cohort and for the subgroups of patients with sPCa found within a blind sector only versus those for whom this criterion was not applicable. The median (interquartile range, IQR) age and PSA of the total cohort was 67.0 (61.0–72.0) years and 8.8 (6.0–12.4) ng/mL, respectively. A total of 400 (36.9%) patients had a history of previous biopsy, 149 (13.7%) underwent biopsy within the scope of an active surveillance strategy. The median total core number was 35.0 (31.0–40.0) with 5.0 (3.0–6.25) target cores and 31.0 (26.0–33.0) random cores. A comparison between the group of patients with sPCa solely localized within a blind sector and patients for whom this criterion was not applicable showed significant differences in the total number of cores taken: 34.0 (28.0–36.0) vs. 35.0 (31.0–40.0) (*p* = 0.04). Moreover, the number of random cores was significantly lower in the group with missed sPCa: 26.0 (24.0–31.0) vs. 31.0 (26.0–33.0) (*p* < 0.01). A lower number of random cores leads to an upsizing of blind sectors and thus results in a higher chance of missing sPCa within a blind sector.

### 3.3. Characterization of Prostate Cancer Missed by the Ginsburg Scheme 

Table 2 presents the localization and classification of PCa that was diagnosed solely or upgraded to sPCa by MR-TB on a patient-level. Significant PCa was missed by the GBS in 39 patients (39/1084, 3.6%). In 19 cases, sPCa was solely diagnosed by MR-TB and 20 cases were upgraded from insignificant to significant by MR-TB. In 27 cases (27/39, 69.2%), sPCa was missed within one of the newly defined blind sectors with 17 lesions localized within the AR, 5 within CTZ, 4 within BD and one in BV. Based on the chi-squared test, the distribution of missed cancerous lesions across the blind sectors was not found to be equal (*p* < 0.001). In seven patients, nsPCa was diagnosed by MR-TB only and thus missed by the GBS. In five of these cases (62.5%), nsPCa was localized within a non-blind sector. Additional sPCa was significantly more often localized in a blind sector, whereas nsPCa was localized more often in a non-blind sector (*p* = 0.04).

Figure 2 shows lesion-level analysis by illustrating both significant (2A) and non-significant (2B) cancerous target lesions that were missed or underestimated by the GBS. Two patients had two cancerous lesions each diagnosed by MR-TB only, resulting in a total of 48 lesions that were missed or underestimated by the GBS. Figure 2A presents the distribution of sPCa according to the newly defined sectors. The majority of missed sPCa was localized within the anterior region (17/40, 42.5%). Figure 2B shows the distribution of nsPCa, with the majority localized within a non-blind sector (5/8 lesions, 62.5%).

### 3.4. Regression Analysis for Predictors of Missing sPCa

The results of univariate and multivariate regression analysis for predictors of missing sPCa within a blind sector of the GBS and within the AR specifically are shown in Table 3. In univariate regression analysis, a higher number of random cores taken according to the GBS significantly decreased the odds of missing cancer within a blind sector and within the AR specifically (Odds ratio (OR) 0.90, 95% confidence interval (CI) 0.85–0.97, *p* < 0.01 and OR 0.90, 95% CI 0.83–0.98, *p* = 0.01, respectively). These findings were consistent in multivariate analysis (OR 0.90, 95% CI 0.83–0.97, *p* < 0.01 and OR 0.91, 95% CI 0.84–1.00, *p* = 0.04, respectively). Applying stepwise regression starting with the full multivariate model, we found a higher number of target cores associated with an increase of missed sPCa within the AR: OR 1.12, 95% CI 1.01–1.24, *p* = 0.03). Notably, the previous biopsy status and prostate volume did not show any significant predictive character.

### 3.5. Prostatic Configuration Analysis

The median PCAR index of the group with sPCa missed within the AR and the matched cohort was 0.71 and 0.75, respectively. This difference was not statistically significant (*p* = 0.40).

## 4. Discussion

Widespread adoption of TP offers the benefit of restricting the use of prophylactic antibiotics and reducing post-TR sepsis rates, both of which translate into a higher procedural safety profile. The GBS has been proposed as a standardized TP biopsy template; however, it mainly focuses on the peripheral zone and inherently omits particular sectors of the prostate. Therefore, adding additional biopsy cores to these blind sectors to detect sPCa may be considered, for example, in patients suspected of harboring PCa who cannot undergo MRI.

In this study, we present the results of a retrospective evaluation of the GBS in terms of a distributional analysis of missed and underestimated PCa by the re-localization of targeted lesions onto a newly defined prostate map. Secondly, we aimed to identify both clinical and anatomical factors that were associated with missing or underestimating sPCa by the GBS and specifically in one of the newly defined blind sectors.

Retrospective analysis of our single-center cohort revealed that sPCa was missed or underestimated by the GBS in a total of 39/1084 patients (3.9%). This rate is comparable to that reported by Radtke and coworkers who analyzed index lesion detection rates by different biopsy strategies in 120 men with a combined biopsy strategy (GBS missed 4.3% of sPCa) and whole-mount histopathology (GBS missed 8.3% of sPCa) as reference standards [16]. In a prospective multi-center study including 487 men, SB according to GBS would have missed sPCa that was found by MR-TB in 12 men (2.5%) [9]. Altogether these results suggest that only a small number of sPCa lesions are missed by the GBS. The number of missed PCa might be potentially higher when referring to whole-mount histopathology as the reference standard. This emphasizes the importance of investigating the anatomical distribution of cancerous lesions that are missed by the GBS.

We subsequently evaluated stored biopsy plans concerning the localization of missed or underestimated sPCa lesions. More than two-thirds were localized within one of the newly defined blind sectors with the majority found within the anterior region. Importantly, the distribution across sectors was found not to be equal with high significance. We identified one study that described the distribution of PCa that was missed by SB [17]. Considering that this analysis referred to a 12-core systematic TRUS-biopsy scheme, it is, therefore, not directly comparable to our analysis of the GBS.

Furthermore, we aimed to identify a subgroup of men that would benefit from adjusting the GBS by adding cores to blind sectors and specifically within the AR. In regression analysis and prostatic configuration analysis, no clinical parameter or anatomical configuration was found to be predictive for sPCa localized within any blind sector or within the AR specifically. Despite the absence of predictors for missing sPCa within a blind sector of the GBS, it is important to consider blind sectors within the scheme, for example, when performing biopsies in patients who cannot undergo MRI or in men with previous negative biopsies. Due to the proximity of the AR to Santorini’s plexus, an increased risk of bleeding-associated complications must be considered when systematically adding cores to this region.

There were several limitations to the current study that must be acknowledged. Even though our study cohort comprised 1084 patients, the number of patients with sPCa that was missed or underestimated by the GBS was relatively small. Consequently, the significance of our distributional analysis of missed cancerous lesions was limited and the regression analysis was not able to identify clinical or anatomical predictors for sPCa localized within a blind sector. Moreover, our retrospective study included biopsy procedures performed by several surgeons with each possibly applying slight individual variations to the GBS. Lastly, the reference standard for the evaluation of the GBS was restricted to the histopathological results of a combined biopsy strategy and validation with whole-mount histopathology was not performed. As a result, lesions that were missed by both MR-TB and SB according to the GBS were lost to our analysis. Considering the effort of analyzing a cohort with a comparable number of radical prostatectomy specimens and that missed lesions defined by the reference standard based on a combined biopsy strategy are truly missed, the methodology applied remains a suitable approach.

To the best of our knowledge, this is the first study to evaluate the distribution of sPCa that was missed or underestimated by the GBS. By adding blind sectors to the GBS, we designed a new sector map of the prostate that can be used for highly accurate reporting of the histopathological results of a combined biopsy strategy (Figure 3). Due to its design derived from the GBS, it is more suitable for this purpose compared to other sector maps either based on geometrical partitioning or evolved for radiological reporting [5,18], respectively [19]. Precise histopathological mapping is crucial for therapy planning, follow-up of patients under active surveillance and in the development of artificial intelligence algorithms for automated sPCa detection and characterization [20].

## 5. Conclusions

Considering a combined biopsy strategy as the reference standard, SB according to the GBS missed or underestimated sPCa in few men only. This is the first study to analyze the distribution of missed cancerous lesions. Seventy percent of sPCa that was missed or underestimated by the GBS was localized within one of the newly defined blind sectors and the majority was found within the anterior region. No clinical or anatomical predictors were identified for sPCa being missed or underestimated within a blind sector. Nevertheless, awareness of blind GBS sectors is crucial, especially when planning biopsies in patients who cannot undergo MRI or when MRI does not reveal any suspicious lesions. Additionally, by incorporating blind sectors to the GBS, we defined a new sector map of the prostate suited for the reporting of histopathological biopsy results.

## Figures and Tables

**Figure 1 cancers-13-02502-f001:**
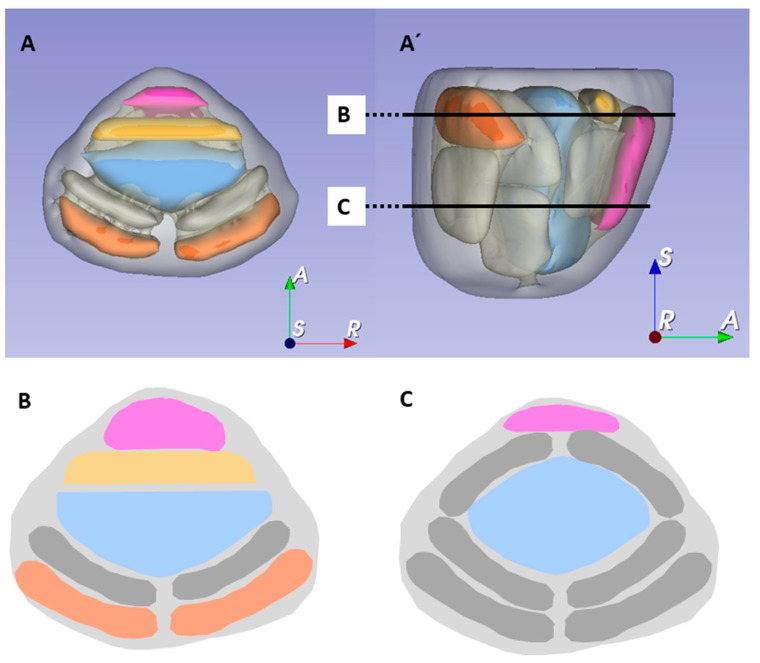
3D visualization of the Ginsburg scheme in different angles (**A**,**A′**). (**B**) and (**C**) show axial planes in two different levels. Inherent blind sectors are marked in colors: anterior region (pink), central transition zone (blue), basoventral sector (yellow), basodorsal sector (orange). The grey-colored sectors illustrate the original Ginsburg sectors.

**Figure 2 cancers-13-02502-f002:**
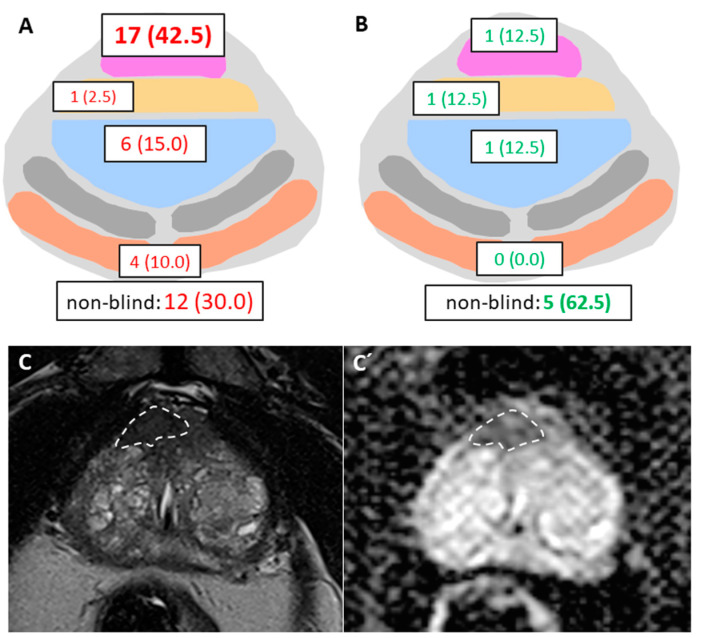
(**A**,**B**) Lesion-based visualization of prostate cancer that was missed or underestimated by systematic biopsy according to Ginsburg scheme in a representative axial plane with all four blind sectors (anterior region in pink, central transition zone in blue, basoventral sector yellow and basodorsal sector orange). (**A**) Localization of sPCa, with the majority localized in the anterior region. (**B**) Localization of nsPCa with the majority localized within a non-blind sector. (**C**,**C′**) MRI with a 15 × 8 mm suspicious lesion in the anterior region with hypointense signal in T2 (**C**) and with a markedly low ADC (**C′**) value. Targeted biopsy diagnosed sPCa (Gleason 6, cancer core length 11 mm).

**Figure 3 cancers-13-02502-f003:**
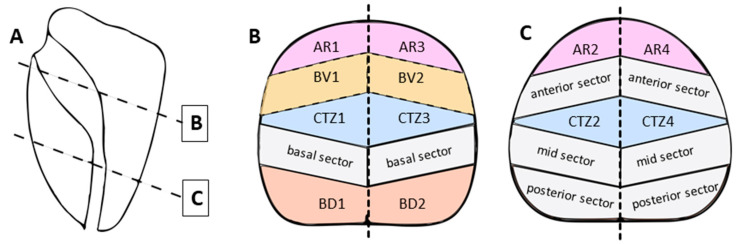
New sector map of the prostate. **A** Sagittal section of the prostate with indication of transversal planes for **B** and **C**. Ginsburg sectors are marked in grey color and nomenclature is in accordance with the pre-established GBS. Four blind sectors are added: anterior region AR1–AR4 (pink), central transition zone CTZ1–CTZ4 (blue), basoventral sector BV1 and BV2 (yellow), and basodorsal sector BD1 and BD2 (orange).

**Table 1 cancers-13-02502-t001:** Patient characteristics between the group with sPCa within blind sector only and no sPCa within blind sector only.

Characteristic	Total	sPCa in Blind Sector Only	No sPCa in Blind Sector Only	*p*-Value
Cases, *n*	1084	27	1057	
Age (years), median, IQR	67.0 (61.0–72.0)	68.0 (61.0–72.5)	67.0 (61.0–72.0)	*p* = 0.78
Previous Biopsy, *n* (%)	400 (36.9)	10 (37.0)	390 (36.9)	*p* = 1.00
Active Surveillance, *n* (%)	149 (13.7)	2 (7.4)	147 (13.9)	*p* = 0.57
PSA (ng/mL), median, IQR	8.8 (6.0–12.4)	7.2 (5.7–13.2)	8.8 (6.0–12.4)	*p* = 0.68
Volume (mL), median, IQR	53.0 (38.4–75.0)	50.3 (34.9–61.1)	53.2 (38.6–75.6)	*p* = 0.09
Number of Cores				
total, median, IQR	35.0 (31.0–40.0)	34.0 (28.0–36.0)	35.0 (31.0–40.0)	*p* = 0.04 *
random, median, IQR	31.0 (26.0–33.0)	26.0 (24.0–31.0)	31.0 (26.0–33.0)	*p* < 0.01 *
target, median, IQR	5.0 (3.0–6.25)	5.0 (3.0–8.0)	5.0 (3.0–6.0)	*p* = 0.89
PI-RADS, *n* (%)				*p* = 0.48
n/a	80 (7.4)	1 (3.7)	79 (7.5)	
1	1 (0.1)	0 (0)	1 (0.1)	
2	45 (4.2)	1 (3.7)	44 (4.2)	
3	175 (16.1)	3 (11.1)	172 (16.3)	
4	545 (50.3)	12 (44.4)	533 (50.4)	
5	238 (22.0)	10 (37.0)	228 (21.6)	
Cancer Detection Rate of Combined Biopsy—ISUP, *n* (%)				
no cancer	411 (37.9)	n/a	411 (38.9)	
1	137 (12.6)	2 (7.4)	135 (12.8)	
2	181 (16.7)	19 (70.4)	162 (15.3)	
3	148 (13.7)	2 (7.4)	146 (13.8)	
4	163 (15.0)	2 (7.4	161 (15.2)	
5	44 (4.1)	2 (7.4)	42 (4.0)	

IQR, interquartile range; PSA, prostate specific antigen; PI-RADS, Prostate Imaging Reporting and Data System; ISUP, classification of the International Society of Urological Pathology; * indicates statistical significance.

**Table 2 cancers-13-02502-t002:** Patient-based analysis of localization and classification of PCa that was diagnosed solely or upgraded to sPCa by targeted biopsy.

Sector	sPCa (Total)	New sPCa	Upgrading to sPCa	Additional nsPCa
overall	39 (100.0%)	19 (48.7%)	20 (51.3%)	7 (100.0%)
non-blind	12 (30.7%) *	5 (12.8%)	7 (17.9%)	5 (71.4%) **
any blind	27 (69.2%) *	14 (35.9%)	13 (33.3%)	2 (28.6%)
AR	17 (43.5%)	9 (23.1%)	8 (20.5%)	1 (14.3%)
CTZ	5 (12.8%)	2 (5.1%)	3 (7.7%)	1 (14.3%)
BD	4 (10.3%)	2 (5.1%)	2 (5.1%)	0 (0%)
BV	1 (2.6%)	1 (2.6%)	0 (0%)	0 (0%)

* Percentages in columns 2–4 refer to the entirety of sPCa (*n* = 39), ** percentages in column 5 refers to the entirety of nsPCa (*n* = 7). AR, anterior region; CTZ, central transition zone; BD, basodorsal; BV, basoventral.

**Table 3 cancers-13-02502-t003:** Univariate and multivariate logistic regression analysis for missing sPCa within any blind sector (**A**) and specifically within the anterior region (**B**).

	(A) Missing sPCa within Any Blind Sector				(B) Missing sPCa in the Anterior Region			
Parameter	Univariate Analysis		Multivariate Analysis ^†^		Univariate Analysis		Multivariate Analysis ^†^	
	OR (95% CI)	*p*	OR (95% CI)	*p*	OR (95% CI)	*p*	OR (95% CI)	*p*
Age, years	1.00 (0.95–1.05)	0.98	1.00 (0.95–1.05)	0.98	1.02 (0.96–1.09)	0.50	1.02 (0.95–1.09)	0.61
Previous Biopsy								
No	Ref		Ref		Ref		Ref	
Yes	1.05 (0.47–2.39)	0.90	1.47 (0.59–3.70)	0.41	1.09 (0.40–2.96)	0.87	0.90 (0.30–2.68)	0.84
Active Surveillance								
No	Ref		Ref		Ref		Ref	
Yes	0.39 (0.05–2.88)	0.35	0.21 (0.02–1.75)	0.15	N/A ^§^		N/A ^§^	
PSA level, ng/mL	0.98 (0.93–1.04)	0.51	0.98 (0.93–1.03)	0.35	0.99 (0.94–1.05)	0.82	0.99 (0.94–1.04)	0.58
Prostate volume, mL	0.99 (0.98–1.01)	0.20	1.00 (0.98–1.01)	0.81	0.99 (0.97–1.01)	0.29	1.00 (0.98–1.02)	0.80
PI-RADS								
1 or 2	Ref		Ref		Ref		Ref	
3	0.79 (0.08–7.73)	0.84	0.67 (0.07–6.87)	0.74	0.26 (0.02–4.21)	0.34	0.23 (0.01–3.88)	0.31
4 or 5	1.30 (0.17–9.87)	0.80	0.97 (0.12–7.72)	0.98	0.88 (0.11–6.80)	0.90	0.59 (0.07–4.88)	0.62
Number of cores								
target	1.07 (0.97–1.18)	0.18	1.06 (0.93–1.20)	0.42	1.13 (1.02–1.26)	0.02 *	1.12 (0.97–1.29)	0.12 ^‡^
random	0.90 (0.85–0.97)	<0.01 *	0.90 (0.83–0.97)	<0.01 *	0.90 (0.83–0.98)	0.01 *	0.91 (0.84–1.00)	0.04 *
Number of lesions	1.23 (0.81–1.87)	0.33	1.03 (0.60–1.79)	0.91	1.45 (0.90–2.33)	0.13	1.08 (0.57–2.06)	0.81

OR, Odds ratio; 95% CI, 95%confidence interval; PSA, prostate specific antigen; PI-RADS, Prostate Imaging Reporting and Data System; Ref, reference category for categorical variables; †, for multivariate analysis, the results of the full model are reported; ‡, statistically significant after variable selection; §, variable was excluded due to limited case number; * indicates statistical significance.

## Data Availability

There is no data availability to report.

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
