# Peer review of "Evaluation of the Ginsburg Scheme: Where Is Significant Prostate Cancer Missed?"

_cancers, 2021, doi:10.3390/cancers13102502_

Round 1

Reviewer 1 Report

Sigle et al describe a study where they looked at the prostate biopsy ginsurg scheme and how much this method is missing significant cancer. singificant cancer was missedn in 3.6%, where the majority is in the anterior area (43.6%).

Study design is interesting and sample size is adequate large.

However major comment is on study design:

GBS performance was compared with Targeted additional bx which I dont really understand:

  • Is this an adequate control group for searching missing cancer?
  • would it not make more sense to compare TBx performance to SB (GBS biopsy)?
  • For GBS biopsy performance one would suggest another control group as for example saturation biopsy by barczell zones or a radical prostatecotmy group?

Some comments:

  • Biopsy procedure: Majority of cases done by one surgeon, but eigth different surgeons? Please clarify.
  • Discussion: Is 3.6% missing cancer relevant? Or is it acceptable? Whats your opinion on this? Is it worth to even consider doing something else if you only miss so less significant cancer? Are you sure you did not miss more?

Author Response

Manuscript: cancers-1187376

Dear Editors,

Dear Reviewers,

Please accept for review our revised manuscript entitled, “Evaluation of the Ginsburg scheme: Where is significant prostate cancer missed?”

We appreciate the thoughtful and instructive comments from our reviewers, and we have addressed each one in a point-by-point fashion. Changes have been incorporated to the revised version of the manuscript.

We thank you very much for your time and consideration, and for the opportunity to submit a revised and improved manuscript. We hope that our revised manuscript is now acceptable for publication. However, please let us know if there are any issues or concerns that require further clarification.

Sincerely yours,

Rodrigo Suarez-Ibarrola, MD.      

University of Freiburg – Medical Centre

Faculty of Medicine                                

Department of Urology

Phone: +49 761 270 28520                                                           

Hugstetter Str. 55, 79106 Freiburg, Germany                                   

rodrigo.suarez@uniklinik-freiburg.de                                                  

 -----------

Reviewer 1:

Sigle et al describe a study where they looked at the prostate biopsy ginsurg scheme and how much this method is missing significant cancer. singificant cancer was missed in 3.6%, where the majority is in the anterior area (43.6%).

Study design is interesting and sample size is adequate large.

Reply 1: Thank you for your positive comment. We would firstly like to point out that the study´s primary aim was to investigate the localization of sPCa lesions that were missed by the GBS.

However major comment is on study design:

GBS performance was compared with Targeted additional bx which I dont really understand:

Is this an adequate control group for searching missing cancer?

would it not make more sense to compare TBx performance to SB (GBS biopsy)?

For GBS biopsy performance one would suggest another control group as for example saturation biopsy by barczell zones or a radical prostatecotmy group?

Reply 2:

Thank you very much for your insightful question. Since TB is not restricted to the Ginsburg sectors it is able to detect PCa outside of GBS areas which were labeled blind sectors in our manuscript. For this reason, it was therefore possible to use a combined biopsy strategy (TB+SB according to the GBS) as a reference standard for the evaluation of the GBS. 
We acknowledge that using the histopathological results of a combined biopsy strategy as reference standard does not respect lesions that were missed by both TB and SB. Consequently, our approach underestimates the total number of PCa that was missed by the GBS.

Using a control group with SB according to the Barzell sectors or based on whole-mount specimens from patients who underwent radical prostatectomy may provide a more adequate estimate of the total number of PCa that was missed by the GBS. On the other hand, considering the large sample size, these approaches are also extremely resource- intensive. These considerations are stated as limitations in the discussion part:

“Lastly, the reference standard for the evaluation of the GBS was restricted to the histopathological results of a combined biopsy strategy and validation with whole-mount histopathology was not performed. As a result, lesions that were missed by both MR-TB and SB according to the GBS were lost to our analysis. Considering the effort of analysing a cohort with a comparable number of radical prostatectomy specimens and that undetected lesions defined by the reference standard based on a combined biopsy strategy are truly missed, the methodology applied remains a suitable approach.“

Furthermore, the study´s main aim was not to evaluate how much PCa was missed by the GBS but rather where it was missed in order to possibly adopt random biopsy schemes.

We added changes to the manuscript to further clarify the study´s aims:

  • Simple Summary: “… This is the first study to evaluate the anatomical distribution of cancerous lesions that were missed by the Ginsburg scheme investigate whether prostate cancer is missed by the Ginsburg scheme and to characterize it in case present…”
  • Introduction: “The study’s primary aim was to evaluate where whether significant PCa (sPCa) is missed by the GBS based on the re-localization of cancerous lesions that were detected by MR-TB only. and to re-localize it within newly defined blind sectors. (…)”
  • Discussion: “(…) In this study, we present the results of a retrospective evaluation of the GBS in terms of a distributional analysis missing or underestimating sPCa based on data from a single-center of consecutive patients who underwent combined TP. Secondly, we examined the distribution of missed and underestimated sPCa by the re-localization of targeted lesions onto a newly defined prostate map (…)”
  • Conclusion: “Considering a combined biopsy strategy as the reference standard, SB according to the GBS missed or underestimated sPCa in few men only.”

Some comments:

Biopsy procedure: Majority of cases done by one surgeon, but eight different surgeons? Please clarify.

Reply 3: Thank you for your valuable comment. We changed the order of sentences for clarification:

“Eight different surgeons performed the biopsy procedures that were reviewed for this study. In the majority of cases the same surgeon performed the planning and execution of both MR-TB and SB.”

Discussion: Is 3.6% missing cancer relevant? Or is it acceptable? Whats your opinion on this? Is it worth to even consider doing something else if you only miss so less significant cancer? Are you sure you did not miss more?

Reply 4: As stated in Reply number 2, the absolute number of missed sPCa may be higher than 3.6% when referring to whole-mount radical prostatectomy specimens as the reference standard. Taking this into account, awareness of blind sectors within the GBS and the distribution of cancerous lesions within these areas is crucial in patients who cannot undergo MRI or when the MRI does not reveal any suspicious lesions.

We added the following sentence to the manuscript:

  1. Discussion: “… The number of missed PCa might be potentially higher when referring to whole-mount histopathology as the reference standard. This emphasizes the importance of investigating the anatomical distribution of cancerous lesions that are missed by the GBS…”

Reviewer 2 Report

The authors retrospectively analyzed prostate cancer detection rates in 1084 patients that underwent MR-TB and SB, in order to test the accuracy of the Ginsburg scheme widely used for sPCa diagnosis. They identified that GBS only missed 3.6% of sPCa, and further found a higher number of random cores associated with a decrease of missing cancer within the AR specifically. The strength of the manuscript is the sample size and research method.

However, on the one hand the authors failed to illustrate a new sector map of the prostate suited for reporting histopathological biopsy results by adding blind sectors to GBS. The major conclusion they propose in the manuscript is not supported by their research. On the other hand, the data is not well explained with little to no discussion in the results. Following are minor comments the review have:

  1. Please indicate the sectors of the grey color
  2. please elaborate on effects of the significant differences seen in the total number of cores taken and the number of random cores taken between sPCa and no sPCa in blind sector only groups. Line 156-161 table 1.
  3. Please explain the numbers in Figure 2 A-B. And discuss the figure in detail in the result section.
  4. Where is the data for the last section of the result: "Prostatic configuration analysis"

Author Response

Manuscript: cancers-1187376 

Dear Editors,

Dear Reviewers,

Please accept for review our revised manuscript entitled, “Evaluation of the Ginsburg scheme: Where is significant prostate cancer missed?”

We appreciate the thoughtful and instructive comments from our reviewers, and we have addressed each one in a point-by-point fashion. Changes have been incorporated to the revised version of the manuscript.

We thank you very much for your time and consideration, and for the opportunity to submit a revised and improved manuscript. We hope that our revised manuscript is now acceptable for publication. However, please let us know if there are any issues or concerns that require further clarification.

Sincerely yours,

Rodrigo Suarez-Ibarrola, MD.      

University of Freiburg – Medical Centre

Faculty of Medicine                                

Department of Urology

Phone: +49 761 270 28520                                                           

Hugstetter Str. 55, 79106 Freiburg, Germany                                   

rodrigo.suarez@uniklinik-freiburg.de

 --------                                                                

Reviewer 2:

The authors retrospectively analyzed prostate cancer detection rates in 1084 patients that underwent MR-TB and SB, in order to test the accuracy of the Ginsburg scheme widely used for sPCa diagnosis. They identified that GBS only missed 3.6% of sPCa, and further found a higher number of random cores associated with a decrease of missing cancer within the AR specifically. The strength of the manuscript is the sample size and research method.

However, on the one hand the authors failed to illustrate a new sector map of the prostate suited for reporting histopathological biopsy results by adding blind sectors to GBS.

Reply 5: Thank you for your comment. We added an illustration of the newly defined sector map suited for reporting histopathological results based on a combined biopsy strategy (Figure 3).

The major conclusion they propose in the manuscript is not supported by their research.

On the other hand, the data is not well explained with little to no discussion in the results.

Reply 6: We appreciate your valuable comment.

We added changes to the manuscript to further clarify the study´s aims:

  • Simple Summary: “… This is the first study to evaluate the anatomical distribution of cancerous lesions that were missed by the Ginsburg scheme investigate whether prostate cancer is missed by the Ginsburg scheme and to characterize it in case present…”
  • Introduction: “The study’s primary aim was to evaluate where whether significant PCa (sPCa) is missed by the GBS based on the re-localization of cancerous lesions that were detected by MR-TB only. and to re-localize it within newly defined blind sectors. (…)”
  • Discussion: “(…) In this study, we present the results of a retrospective evaluation of the GBS in terms of a distributional analysis missing or underestimating sPCa based on data from a single-center of consecutive patients who underwent combined TP. Secondly, we examined the distribution of missed and underestimated sPCa by the re-localization of targeted lesions onto a newly defined prostate map (…)”
  • Conclusion: “Considering a combined biopsy strategy as the reference standard, SB according to the GBS missed or underestimated sPCa in few men only.”

Following are minor comments the review have:

  • Please indicate the sectors of the grey color

Reply 7: Thank you for your comment. We added a sentence to the legend of Figure 1:

The grey coloured sectors illustrate the original Ginsburg sectors.”

  • please elaborate on effects of the significant differences seen in the total number of cores taken and the number of random cores taken between sPCa and no sPCa in blind sector only groups. Line 156-161 table 1.

Reply 8: Thank you very much for bringing up this point. Our interpretation of this finding is quite straight-forward: A higher number of random cores downsizes the inherent blind areas of the Ginsburg scheme and thus leads to missing less sPCa within a blind sector.

We added the following sentence to the results section:

“A lower number of random cores lead to an upsizing of blind sectors and thus results in a higher chance of missing sPCa within a blind sector.”

  • Please explain the numbers in Figure 2 A-B. And discuss the figure in detail in the result section.

Reply 9: As this is probably one of the most important sections in our manuscript, we agree that it is a very good idea to provide more detail about Figure 2 in the result section.

We added the following sentences to the manuscript:

Figure 2A presents the distribution of sPCa according to the newly defined sectors. The majority of missed sPCa was localized within the anterior region (17/40, 42.5%). Figure 2B shows the distribution of nsPCa with the majority localizing within a non-blind sector (5/8 lesions, 62.5%).”

  • Where is the data for the last section of the result: "Prostatic configuration analysis"

Reply 10: Since the prostatic configuration analysis data only comprises the medians of the respective groups and the p-value of the Welch´s two sample test, it is solely reported this way in the last part of results section.

Round 2

Reviewer 1 Report

Thanks for explaining my raised questions. I have no further comments

Author Response

Dear reviewer

On the contrary, thank you very much for improving our manuscript with the questions you raised. 

Reviewer 2 Report

Thank the authors for the revised manuscript. All of the reviewer's comments have been addressed properly. 

One last comment is: please color code figure 3 as indicated in the figure legend.

Author Response

Dear reviewer

Thank you very much for improving our manuscript with the questions you kindly raised.

We improved and color-coded Fig 3 as specified in its legend.